
# Differential absorption lidar for water vapor isotopologues in the 1.98 μm spectral region: sensitivity analysis with respect to regional atmospheric variability

Jonas Hamperl[1], Clément Capitaine[2], Jean-Baptiste Dherbecourt[1], Myriam Raybaut[1], Patrick Chazette[3], Julien Totems[3], Bruno Grouiez[2], Laurence Régalia[2], Rosa Santagata[1], Corinne Evesque[4], Jean-Michel Melkonian[1], Antoine Godard[1], Andrew Seidl[5], Harald Sodemann[5], Cyrille Flamant[6]

[1]DPHY, ONERA, Université Paris Saclay, F-91123 Palaiseau, France
[2]Groupe de Spectrométrie moléculaire et atmosphérique (GSMA) UMR 7331, URCA, France
[3]Laboratoire des Sciences du Climat et de l'Environnement (LSCE), UMR 1572, CEA-CNRS-UVSQ, Gif-sur-Yvette, France
[4]Institut Pierre-Simon Laplace (IPSL), FR636, Guyancourt, France
[5]Geophysical Institute, University of Bergen, and Bjerknes Centre for Climate Research, Bergen, Norway
[6]Laboratoire Atmosphères Milieux et Observations Spatiales (LATMOS), UMR 8190, CNRS-SU-UVSQ, Paris, France

*Correspondence to*: Jonas Hamperl (jonas.hamperl@onera.fr)

**Abstract.** Laser active remote sensing of tropospheric water vapor is a promising technology to complement passive observational means in order to enhance our understanding of processes governing the global hydrological cycle. In such context, we investigate the potential of monitoring both water vapor $H_2^{16}O$ and its isotopologue $HD^{16}O$ using a differential absorption lidar (DIAL) allowing for ground-based remote measurements at high spatio-temporal resolution (150 m and 10 min) in the lower troposphere. This paper presents a sensitivity analysis and an error budget for a DIAL system under development which will operate in the two-micrometer spectral region. Using a performance simulator, the sensitivity of the DIAL-retrieved mixing ratios to instrument-specific and environmental parameters is investigated. This numerical study uses different atmospheric conditions ranging from tropical to polar latitudes with realistic aerosol loads. Our simulations show that the measurement of the main isotopologue $H_2^{16}O$ is possible over the first 1.5 km of atmosphere with a relative precision in the water vapor mixing ratio of <1% in a mid-latitude or tropical environment. For the measurement of $HD^{16}O$ mixing ratios under the same conditions, relative precision is shown to be of similar order, thus allowing for the retrieval of range-resolved isotopic ratios. We also show that expected precisions vary by an order of magnitude between tropical and polar conditions, the latter giving rise to reduced precision due to low water vapor content and low aerosol load. Such values have been obtained for a commercial InGaAs PIN photodiode, as well as temporal and line-of-sight resolutions of 10 min and 150 m, respectively. Additionally, using vertical isotopologue profiles derived from a previous field campaign, precision estimates for the $HD^{16}O$ isotopic abundance are provided.



# 1 Introduction

In many important aspects, climate and weather depend on the distribution of water vapor in the atmosphere. Water vapor leads to the largest climate change feedback, as it more than doubles the surface warming from atmospheric carbon dioxide (Stevens et al., 2009). Knowing exactly how water vapor is distributed in the vertical is of paramount importance for understanding the lower-tropospheric circulation, deep convection, the distribution of radiative heating, surface fluxes magnitude and patterns, among other processes. Conventional radio-sounding or passive remote sensors, such as microwave

radiometers or infrared spectrometers, are well established tools used for water vapor profile retrieval in the atmosphere. However, apart from balloon-borne soundings, most of these instruments do not allow for determining how water vapor is distributed along the vertical in the 0–3 km above the surface which contains 80% of the water vapor amount of the atmosphere. Additionally, passive remote sensors will generally require ancillary measurements such as aerosols, temperature, or cloud heights to limit the errors on retrieved concentrations from radiance measurements. To complement

these methods, active remote sensing techniques are expected to provide higher resolution measurement capabilities especially in the vertical direction where the different layers of the atmosphere are directly probed with a high-power laser transmitter. Among these active remote sensing techniques, Raman lidar is a powerful way to probe the atmosphere as it can give access to several atmospheric state parameters within a single line of sight such as temperature, aerosols, and water vapor mixing ratio (WVMR) (Whiteman et al., 1992). Benefiting from widely commercially available high-energy visible or

UV lasers, as well as highly sensitive detectors, they allow high accuracy, long range measurements despite the small Raman scattering cross-section. WVMR retrieval from Raman Lidar signal is however typically limited by parasitic daytime sky radiance and requires instrument constant and overlap function calibration (Whiteman et al., 1992; Wandiger and Raman, 2005). Conversely, the differential absorption lidar (DIAL) technique is in principle calibration free since the targeted molecule mixing ratio can be directly retrieved from the attenuation of the lidar signals at two different wavelengths,

knowing the specific differential absorption cross-section of the targeted molecule (Bösenberg, 2005). However, this benefit must be balanced with higher instrumental constraints especially on the laser source which is required to provide high power as well as high frequency agility and stability at the same time. For water vapor this method has been successfully demonstrated essentially using pulsed laser sources emitting in the visible or near infrared (Bruneau et al., 2001; Wirth et al., 2009; Wagner and Plusquellic, 2018), and recent progress in the fabrication and integration of tapered semiconductor optical

amplifiers has enabled the development of small-footprint field-deployable instrumentation (Spuler et al., 2015). The infrared region between 1.5 µm and 2.0 µm has also attracted interest for water vapor DIAL sounding, especially in the context of co-located methane and carbon dioxide monitoring (Wagner and Plusquellic, 2018, Cadiou et al., 2016). One of the potential benefits of co-located multiple species measurement would be to reduce the uncertainties related to the retrieval of dry-air volume mixing ratios for the greenhouse gas (GHG) of interest. This aspect has particularly been studied in the

field of space-borne integrated path differential absorption (IPDA) lidar for carbon dioxide ($CO_2$) monitoring in the 2.05 µm region where water vapor absorption lines may affect the measurement (Refaat et al., 2015). One of the great potentials of



these multiple-wavelengths and multiple-species approaches would be their adaptability to isotopologue measurements with the DIAL technique since isotopic ratio estimation is equivalent to multiple species measurement provided the targeted isotopologues display similarly suitable and well separated absorption lines in a sufficiently narrow spectral window.


Humidity observations alone are not sufficient for identifying the variety of processes accounting for the proportions and history of tropospheric air masses (Galewsky et al., 2016). Stable water isotopologues, mainly $H_2^{16}O$, $HD^{16}O$ and $H_2^{18}O$ differ by their mass and molecular symmetry. As a result, during water phase transitions, they have slightly different behaviors. The heavier molecules prefer to stay in the liquid or solid phase while the lighter ones tend to evaporate more

easily, or prefer to stay in the vapor phase. This unique characteristic makes water isotopologues the ideal tracers for processes in the global hydrological cycle. Water isotopologues are independent quantities depending on many climate factors, such as vapor source, atmospheric circulation, precipitation and droplet evaporation, and ambient temperature. So far, no lidar system has been investigated for the measurement of water vapor isotopologues other than $H_2^{16}O$ (hereafter referred to as $H_2O$). Here, in the framework of the Water Vapor Isotope Lidar (WaVIL) project (Wavil, 2021), we investigate

the possibility of a transportable differential absorption lidar to measure the concentration of both water vapor $H_2O$ and the isotopologue $HD^{16}O$ (hereafter referred to as HDO) at high spatio-temporal resolution in the lower troposphere (Hamperl et al., 2020). The proposed lidar will operate in the two-micrometer spectral region where water vapor isotopologues display close but distinct absorption lines. Such an innovative remote sensing instrument would allow for the first time the simultaneous monitoring of water vapor and HDO isotopic abundance profiles, enabling the improvement of knowledge on

the water cycle at scales relevant for meteorological and climate studies.

The purpose of this paper is to assess the expected performances of a DIAL instrument for probing of $H_2O$ and HDO in the lower troposphere. In section 2, the choice of the sensing spectral range is substantiated, and the performance model is outlined. The approach for modelling transmitter, detection, and environmental parameters is detailed. The sensitivity

analysis is based on representative average columns of arctic, mid-latitude, and tropic environments. The simulation results and an extensive error analysis are presented in section 3. To assess the random uncertainty in the retrieved isotopologue mixing ratio, major detection noise contributions are analyzed for a commercial InGaAs PIN and a state-of-the-art HgCdTe avalanche photodiode. Instrument- and atmosphere-specific systematic errors are discussed for different model environments. Finally, performance calculations were applied to vertical profiles retrieved from a past experimental

campaign where a Raman lidar for water vapor measurements along with in-situ sensors for the HDO isotopologue measurements were deployed. A conclusion and perspectives for forthcoming calibration and validation field campaigns are given in section 4.


## 2 DIAL method and performance model for water vapor isotopologue measurement

### 2.1 Choice of the sensing spectral range

Remote sensing by DIAL relies on the alternate emission of at least two laser wavelengths, named $\lambda_{on}$ and $\lambda_{off}$, respectively in coincidence with and out of a gas absorption feature, to retrieve a given species concentration. The key to independently measure HDO and $H_2O$ abundances with a single instrument lies thus in the proper selection of a spectral region where: *i)* the two molecules display well separated, significant absorption lines while minimizing the interference from other atmospheric species, and *ii)* the selected lines should preserve relatively equal lidar signal dynamic and relative precision

ranges for both isotopologues. This makes the line selection rather limited. Using spectroscopic data from the HITRAN 2016 database (Gordon et al., 2017), we investigated the possibilities for HDO sounding up to 4 µm, where robust pulsed nanosecond lasers or optical parametric oscillator sources based on mature lasers or nonlinear crystals components can be developed (Godard, 2007). Figure 1a shows that HDO lines are strong in the 2.7 µm region but overlap with an even more dominant $H_2O$ absorption band. Considering the state of possible commercial photodetector technologies, we chose to limit

the range of investigation to 2.6 µm, corresponding to the possibilities offered by InGaAs photodiodes. In the telecom wavelength range, which offers both mature laser sources and photodetectors, HDO absorption lines are too weak to be exploited for DIAL measurements over 1–3 km. The same argumentation holds for wavelengths towards 2.05 µm (see Fig. 1b) which have been extensively studied for space-borne $CO_2$ IPDA lidar sensing (Singh et al., 2017; Ehret et al., 2008). However, the 2 µm region seems to offer an interesting possibility in terms of absorption strength as well as technical

feasibility of pulsed, high-energy, single-frequency laser sources (Geng et al., 2014). The spectral window between 1982–1985 nm is well suited to meet the mentioned requirements as illustrated in Fig. 1c. In this paper we will focus on the lines at the positions 5043.0475 cm$^{-1}$ (1982.93 nm) and 5044.2277 cm$^{-1}$ (1982.47 nm), respectively for $H_2O$ and HDO (hereafter referred to as option 1), allowing for a sufficiently high absorption over several kilometers with negligible interference from other gas species. Additionally, a second option for $H_2O$ slightly detuned from the absorption peak at 1982.97 nm will be

discussed (hereafter referred to as option 2). Wavelength switching will be realized on a shot-to-shot basis to consecutively address the chosen off-line wavelength at 1982.25 nm and the on-line wavelengths for $H_2O$ and HDO. As shown in Fig. 1c, the HDO absorption line is accompanied by a non-negligible $H_2O$ absorption which has to be corrected for when retrieving the volume mixing ratio and thus adding a bias dependent on the accuracy of the $H_2O$ measurement at 1982.93 nm. Alternatively, measurement within the spectral window between 1983.5 nm and 1984.5 nm is also possible for simultaneous

$H_2O$ and HDO probing, however with weaker absorption giving rise to smaller signal-to-noise ratios and consequently increased measurement statistical uncertainty. In any of the proposed cases, addressing the on-line and off-line spectral features requires a tuning capability larger than 0.5 nm which can be offered, for instance, by an optical parametric oscillator source (Cadiou et al., 2016; Barrientos Barria et al., 2014), which is envisioned to be used for the WaVIL system.





## 2.2 DIAL performance model

The objective of the presented performance model is to elaborate the precision achievable with the proposed DIAL instrument of the volume mixing ratios of the water isotopologues $H_2O$ and HDO and thus of the precision on the measurement of HDO abundance (noted $\delta D$) which expresses the excess (or defect) of the deuterated isotope compared to a reference value of $311.5 \cdot 10^{-6}$ (one HDO molecule for 3115 $H_2O$ molecules) (Craig, 1961). Following the convention, the HDO abundance (in permil, ‰) is expressed as the deviation from that of the standard mean ocean water (SMOW) in the so-
called notation:

$$\delta D \ = \ 1000 \ \times \ \left[ \frac{[HDO]_{sample} \ / \ [H_2O]_{sample}}{[HDO]_{SMOW} \ / \ [H_2O]_{SMOW}} \ - \ 1 \right] \tag{1}$$

where [ ] represents the concentration of $H_2O$ and HDO.

As schematically depicted in Fig. 2, the DIAL simulator consists of three sub-models describing atmospheric properties, lidar instrument parameters, and detector properties. Each model will be explained in a more details in the following
paragraphs. The atmosphere model is based on a set of standard profiles of temperature, pressure, and humidity representative of different climate regions along with aerosol optical depth data of the AERONET database. Those data are exploited to calculate the atmospheric transmission using absorption cross-sections computed with the HITRAN2016 spectroscopic database (Gordon et al., 2017). Together with the model describing the lidar instrument, the calculated transmission data are used to feed the lidar equation in order to calculate the received power at each selected on-line and off-
line wavelength. In a subsequent step, noise contributions arising from the detection unit are taken into account to estimate the signal-to-noise ratio. Then, we use an analytical approach based on an error propagation calculation to estimate the random error on the measured isotopic mixing ratios and thus the uncertainty of the $\delta D$ retrieval obtained with the simulated instrumental parameters.

Starting from the lidar equation (Collis and Russell, 1976), the calculated received power as a function of distance $r$ writes as:

$$P_r(r) \ = \ T_r \frac{A}{r^2} \ \beta_\pi(r) \ O(r) \ \frac{c}{2} \ T_{atm}^2(r) \ E_p \tag{2}$$

where $T_r$ is the receiver transmission, $A$ is the effective area of the receiving telescope, $\beta_\pi(r)$ is the backscatter coefficient, $O(r)$ is the overlap function between the laser beam and the field of view of the receiving telescope, $c$ is the speed of light,
$T_{atm}(r)$ is the one-way atmospheric transmission and $E_p$ the laser pulse energy. The DIAL technique is based on the emission of two wavelengths in ($\lambda_{on}$) and out of ($\lambda_{off}$) coincidence with a target gas absorption line. Provided that the two laser pulses are emitted sufficiently close in time for the atmospheric aerosol content to be equivalent, the two wavelengths experience





the same backscattering along the line of sight, and the differential optical depth $\Delta\tau$ as the difference of on- and off-line optical depth at a measurement range $r$ can be retrieved by:

$$\Delta\tau(r) = \frac{1}{2}\ln\left(\frac{P_{\text{off}}(r)}{P_{\text{on}}(r)}\right) \tag{3}$$


with $P_{\text{on}}$ and $P_{\text{off}}$ as the backscattered power signals for $\lambda_{\text{on}}$ and $\lambda_{\text{off}}$, respectively. Using the optical depth measurement, the gas concentration can be retrieved at a remote range $r$ within a range cell $\Delta r = r_2 - r_1$. Assuming $\Delta r$ is sufficiently small, the water vapor content expressed as volume mixing ratio, which is assumed as constant within $\Delta r$, can then be derived by:

$$X_{\text{H2O}}(r_1 \rightarrow r_2) = \frac{\Delta\tau(r_2) - \Delta\tau(r_1)}{\int_{r_1}^{r_2} WF(r)dr} \tag{4}$$

with $WF(r)$ representing a weighting function defined as:

$$WF(r) = \left(\sigma_{\text{on}}(r) - \sigma_{\text{off}}(r)\right)\rho_{\text{air}}(r) \tag{5}$$

where $\rho_{\text{air}}$ is the total air number density and $\sigma_{\text{on}}$ and $\sigma_{\text{off}}$ are the on-line and off-line absorption cross-sections calculated with the HITRAN 2016 spectroscopic database assuming a Voigt profile. The given formulas are valid for the detection of the main isotopologue $H_2O$. For HDO however, the presence of $H_2O$ absorption at the on-line wavelength of HDO (see Fig. 1c)

necessitates an additional consideration of that bias for the inversion. Taking this into account, Eq. (3) changes to:

$$\Delta\tau_{\text{HDO}}(r) = \frac{1}{2}\ln\left(\frac{P_{\text{off}}(r)}{P_{\text{on}}(r)}\right) - \Delta\tau_{\text{H2O}}(r) \tag{6}$$

where $\Delta\tau_{\text{H2O}}$ represents the $H_2O$ differential optical depth at the HDO on-line wavelength $\lambda_{\text{HDO}}$ which can be calculated with the knowledge of the volume mixing ratio $X_{\text{H2O}}$ measured at $\lambda_{\text{H2O}}$.

To obtain an analytical expression for the random error in the concentration measurement, an error propagation calculation

can be applied to Eqs. (3) and (4) assuming that the range cell interval $\Delta r$ is sufficiently small and that the range cell resolution of the receiver is sufficiently high to consider $\Delta\tau(r_1)$ and $\Delta\tau(r_2)$ as uncorrelated. The absolute uncertainty in the volume mixing ratio $X$ expressed as standard deviation $\sigma(X)$ can be calculated from the signal-to-noise ratios of the on- and off-line power signals as follows:

$$\sigma(X) = \frac{\Delta f}{\sqrt{2}\ WF\ c}\left(\frac{1}{SNR_{\text{on}}^2} + \frac{1}{SNR_{\text{off}}^2}\right)^{1/2} \tag{7}$$

where $\Delta f$ is the measurement bandwidth which is the same for the on- and off-line pulses since they are measured sequentially by the same detector. Finally, with both uncertainties in the volume mixing ratios $X_{\text{H2O}}$ and $X_{\text{HDO}}$ known, an estimation of the uncertainty in $\delta D$ is obtained by applying an error propagation calculation to Eq. (1) in order to get the expected uncertainty expressed as variance:





$$\sigma(\delta D) \; = \; (\delta D + 1) \left\{ \left( \frac{\sigma(X_{H2O})}{X_{H2O}} \right)^2 + \left( \frac{\sigma(X_{HDO})}{X_{HDO}} \right)^2 \right\}^{1/2} \tag{8}$$

## 2.3 Instrument and detector model

In order to estimate the feasibility of a DIAL measurement, calculations were performed for the transmitter and receiver parameters summarized in Table 1. The emitter of the DIAL system will be based on a generic optical parametric oscillator/optical parametric amplifier (OPO/OPA) architecture as the one developed in (Barrientos Barria et al., 2014). The combination of a doubly-resonant Nested Cavity OPO (NesCOPO) and an OPA pumped by a 1064 nm Nd:YAG commercial laser with 150 Hz repetition rate allows for single-frequency, high-energy pulses with adequate tunability. From this system we expect an extracted signal energy of up to 20 mJ at 1983 nm. For a more conservative estimate, we will also consider a lower-limit pulse energy of 10 mJ for our simulations. The receiver part consists of a Cassegrain-type telescope with a primary mirror of 40 cm in diameter. For the detection part, calculations were performed in a direct-detection setup for i) a commercial InGaAs PIN photodiode and ii) a HgCdTe avalanche photodiode (APD) specifically developed for DIAL applications in the 2 µm range, presented in (Gibert et al., 2018). Given the small active area of the APD, aligning the optics and the imaging of the field of view on the detector might prove extremely challenging in practice. However, for our simulations we do not take this into account and assume the same imaging optics for both the PIN photodiode and the APD except for a reduced diaphragm diameter for the APD, thus resulting in different field-of-view angles. The measurement bandwidth of the DIAL system is effectively determined by an electronic low-pass filter in the detection chain. In the simulation we use a bandwidth setting of 1 MHz corresponding to a spatial resolution of the retrieved isotopologue concentrations of 150 m. For all our calculations we assume signal averaging over an integration time of 10 min (30 000 laser shots for each wavelength).

In order to quantify the measurement uncertainty in the retrieved isotope mixing ratios, random and systematic sources of errors are taken into account. Random errors in measuring the differential optical depth, and thus the species mixing ratio, are related to different noise contributions arising from the detection setups. For a single return-signal pulse, the associated noise power $P_n$ consists of a constant detector and amplifier noise expressed as noise equivalent power $NEP$, shot noise due to background radiation $P_{sky}$, shot noise dependent on the pulse power $P(\lambda)$, as well as speckle noise $P_{sp}(\lambda)$:

$$P_n \; = \; \sqrt{\left( NEP^2 + 2 \cdot e \cdot \left[ P_{sky} + P(\lambda) \right] \cdot F \; / \; R \right) \cdot \Delta f \; + \; P_{sp}^2(\lambda)} \tag{9}$$

where $e$ is the elementary charge, $F$ the excess noise factor (in case of the APD), $R$ the detector responsivity (depending on quantum efficiency in case of the APD) and $\Delta f$ the measurement bandwidth. The $NEP$ of 600 fW Hz$^{-1/2}$ for configuration i) featuring the InGaAs PIN photodiode is a conservative estimate by calculations based on the specifications of the photodiode





and amplifier manufacturer (G12182-003K InGaAs PIN photodiode from Hamamatsu combined with a gain adjustable DHPCA-100 current amplifier from FEMTO). The background power $P_{sky}$ depends on the background irradiance $S_{sky}$ and

the receiver geometry according to:

$$P_{sky} = \frac{\pi}{4} \cdot S_{sky} \cdot \Delta\lambda_f \cdot A_{eff} \cdot \theta_{FOV}^2 \tag{10}$$

where $\Delta\lambda_f$, $A_{eff}$ and $\theta_{FOV}$ are the optical filter bandwidth, effective receiver telescope area and field of view angle, respectively. Assuming Gaussian beam characteristics, the speckle-related noise power is approximately given by (Ehret et al., 2008):

$$P_{sp} = P(\lambda) \cdot \frac{\lambda \cdot 2\sqrt{\Delta f \cdot \tau_c}}{\pi \cdot R_{tel} \cdot \theta_{FOV}} \tag{11}$$

where $R_{tel}$ denotes the telescope radius and $\tau_c$ the coherence time of the laser pulse corresponding to the pulse duration for a Fourier-transform-limited pulse. Finally, the overall time-averaged signal-to-noise ratio is given as the ratio of received power from Eq. (2) and total noise power from Eq. (9) multiplied by the square root of the number of laser shots $N$:

$$SNR = \frac{P_r}{P_n} \sqrt{N} \tag{12}$$

**2.4 Atmosphere model**

We constructed different atmospheric models for mid-latitude, arctic, and tropical locations to study the sensitivity of the DIAL measurement to environmental factors. The atmosphere model consists of vertical profiles of pressure, temperature and humidity (see appendix for origin of sounding data) which serve as input to calculate altitude-dependent absorption cross-sections using the HITRAN 2016 spectroscopic database. For the sake of simplicity, HDO mixing ratios were obtained

from $H_2O$ profiles simply by considering their natural abundance of $3.11 \cdot 10^{-4}$, i.e., variability in terms of the isotopic ratio $\delta D$ is not assumed in our model. For each location, a baseline model was constructed by using the columns of pressure, temperature and volume mixing ratios averaged over the year of 2019. To reflect seasonal variations in our sensitivity analysis, we use profiles with the lowest and highest monthly averages of temperature and humidity (Fig. 3 a–c). To complement the atmospheric model, data of level 2 aerosol optical depth (AOD) from the AERONET database

(https://aeronet.gsfc.nasa.gov/) were used. AERONET sun photometer products are usually available for wavelengths between 340 nm and 1640 nm. For extrapolation to the 2 μm spectral region, we used the wavelength dependence of the AOD described by a power law of the form (Angström, 1929):

$$\frac{AOD(\lambda)}{AOD(\lambda_0)} = \left(\frac{\lambda}{\lambda_0}\right)^{-\alpha} \tag{13}$$

where $AOD(\lambda)$ is the optical depth at wavelength $\lambda$, $AOD(\lambda_0)$ is the optical depth at a reference wavelength, and $\alpha$ represents

the Angstrom exponent. The Angstrom exponent was obtained by fitting Eq. (13) to the available AOD data in the above-





mentioned spectral range in order to extrapolate further to 1.98 µm. Histograms of the yearly distribution of the extrapolated AOD at 1.98 µm are shown in the right column of Fig. 3 (g–i). Median values of the AOD are used for the baseline model. The lowest ($AOD_{10}$) and highest ($AOD_{90}$) decile values serve as input for the sensitivity analysis to model conditions of low and high aerosol charge, respectively. As a next step, vertical profiles of aerosol extinction are constructed by making basic
assumptions about their shape and constraining their values by the extrapolated AOD. In our baseline model, the vertical distribution of aerosols is represented by an altitude-dependent Gaussian profile of the extinction coefficient with varying half-width depending on the location (Fig. 3 d–f). This type of profile roughly corresponds to the ESA Aerosol Reference Model of the Atmosphere (ARMA) (ARMA, 1999) which is plotted for each region normalized to the $AOD_{90}$-derived extinction profile maximum.

However, the distribution of tropospheric aerosols varies widely from region to region (Winker et al., 2013). To broadly reflect the different boundary layer characteristics for each environment, the extinction profile was adapted accordingly. In mid-latitude regions, vertical aerosol distributions vary widely due regional and seasonal factors (Chazette and Royer, 2017). The PBL height can range from a few hundred meters up to 3 km (Matthias et al., 2004; Chazette et al., 2017). Assuming that aerosols are mostly confined to the PBL and that the free-tropospheric contribution to aerosol extinction is weak, the
half-Gaussian-shaped baseline model used for the simulations gives rise to 85% of AOD within the first 1.5 km. Since high aerosol loads in the free troposphere due to long-range dust transport are not uncommon over Western Europe (Ansmann et al., 2003), a dust scenario profile constrained by the highest-decile AOD was also investigated. Dust aerosols are represented by a Gaussian profile above the PBL extending well up to a height of 5 km. For this case, aerosol extinction in the PBL below 1.5 km accounts for half of the total AOD, while dust in the free troposphere accounts for the other half. At high
latitudes, the boundary layer tends to be stable and extends from a few meters to a few hundred meters above ground. Our baseline Arctic extinction profile thus contains 95% of the AOD within the first 1.5 km since most aerosols are confined within the first kilometer of the troposphere as observed by space-born lidar during long-term studies of the global aerosol distribution (Di Pierro et al.,2013). The occurrence histogram in Fig. 3h shows very low values of AOD for most of the time in the available photometer products from February to September. The long-tailed wing of the asymmetric distribution
towards higher values can be explained by seasonally occurring episodes of arctic haze due to anthropogenic aerosols transported from mid-latitude regions (winter to spring) and boreal forest fire smoke during the summer season (Tomasi et al., 2015; Chazette et al., 2018). Similar to the dust scenario for the mid-latitude model, haze and smoke events are modelled by an additional Gaussian profile in the free troposphere constrained by the highest-decile AOD. Extinction profiles representing the tropical environment of La Réunion Island, where sea salt aerosols can be assumed to be the dominant
aerosol species, are chosen such that 90% of the AOD is contributed to the first 1.5 km.

Vertical profiles of the aerosol backscatter coefficient were calculated assuming, for the sake of simplicity, a constant extinction-to-backscatter ratio (lidar ratio) of 40 sr throughout all sets of extinction profiles.



## 3 Simulation results and discussion

### 3.1 Instrument random error

This section aims to quantify the random error on the mixing ratio measurement depending on instrument settings such as laser pulse energy and the type of detector employed. All calculations are based on the mid-latitude baseline atmosphere model assuming vertical sounding of the lower troposphere with aerosols confined to the lowest 2 km. Considering a simple calculation of random errors we will discuss their implications on the precision of the measurement of range-resolved $\delta D$ profiles. Given the instrument parameters presented in Table 1, the dominant noise contributions are estimated which are

shown for a single on-line pulse in Fig. 4 for both detector configurations.

As expected, the overall noise level is significantly reduced by roughly one order of magnitude for the HgCdTe APD combined with a transimpedance amplifier due to a low combined NEP of 75 fW Hz$^{-1/2}$ compared to 600 fW Hz$^{-1/2}$ for the amplifier of the InGaAs PIN detector. In fact, shot noise and electronic noise are at the same level for the APD for a height up to 1 km whereas the electronic noise of the transimpedance amplifier is the predominant contribution over the entire

range for the commercial PIN detector. Signal-to-noise ratios up to $10^2$ are obtained for a single measurement pulse within the first kilometer. Integrating over 30 000 laser shots (equivalent to 10 min averaging time if the two on-line and one off-line wavelengths are addressed sequentially) would increase the signal-to-noise ratios to over $10^4$ in the first kilometer for both detectors and to values around $10^2$ at a 2 km range for the commercial PIN detector and $10^3$ for the HgCdTe APD.

The expected relative random errors on the mixing ratios of $H_2O$ and HDO are shown separately in Figs. 5a and 5b for each detector. We examined two scenarios with different laser pulse energies of 10 mJ and 20 mJ, a measurement bandwidth of 1 MHz (150 m range cell resolution) and an integrating time of 10 minutes for a repetition rate (on-off rate) of 150 Hz. The simulation based on the 20 mJ configuration gives an estimation of the precision limit of the DIAL system. The second configuration with 10 mJ pulse energy can be understood as a lower limit on the precision of measuring mixing ratios of

$H_2O$ and HDO, and finally $\delta D$. As shown in Fig. 5a, a relative random error of well under 1% on the mixing ratio of both $H_2O$ and HDO can be achieved within the first kilometer for both detectors and 20 mJ pulse energy. The degraded precision for measuring HDO is due to its lower differential absorption compared to $H_2O$. For the low-noise APD shown in Fig. 5b, the simulations show that even for the conservative assumption of 10 mJ pulse energy, the relative error stays below 1% for both $H_2O$ and HDO over a range of 1.5 km corresponding to typical heights of the planetary boundary layer. The simulation

results also reveal a sharp rise in the random uncertainty towards longer distances which is attributed to the drastic decline of aerosol backscatter in the free troposphere in our model. The sharp fall of the random error within the first 200–300 m is due to the increasing overlap between laser beam and telescope field of view imaged onto the detector described by the overlap function $O(r)$ in Eq. (2). This overlap term is zero right in front of the lidar instrument and reaches unity after a few hundred meters. It should be noted that $H_2O$ uncertainties were calculated for sounding at the peak of the absorption line (option 1).





Figure 5c shows the expected precision in $\delta D$ which depends on the relative random errors of the volume mixing ratios for $H_2O$ and HDO (see Eq. (7)). For the commercial InGaAs PIN photodiode we find for the limiting case of high measurement precision (20 mJ pulse energy, 1 MHz bandwidth) that the absolute value of uncertainty in $\delta D$ is below 3‰ within a range of 1 km. The 10 mJ configuration also allows for measurement of $\delta D$, however with deteriorated absolute precision up to 10‰ within the first kilometer. Simulations with the HgCdTe APD detector indicate that an absolute precision level lower than

10‰ within the first 1.5 km can be achievable.

### 3.2 Sensitivity to atmospheric variability

The sensitivity of the DIAL instrument to the variability in temperature, humidity, and aerosol load was investigated for the mid-latitude, arctic and tropical atmosphere models. In the following analysis, the relative random error (precision) is used to compare the influence of each atmospheric parameter under investigation. Simulation results are summarized in Fig. 6 for

targeting $H_2O$ (blue) and HDO (red). Here again, we consider a measurement bandwidth of 1 MHz (150 m range cell resolution), and an integrating time of 10 minutes for a repetition rate of 150 Hz. All calculations have been performed with the InGaAs PIN detector and assuming a laser pulse energy of 20 mJ.

Starting with temperature, no effect on the measurement random error was found when simulating under conditions of lower and higher temperature compared to the average atmospheric columns. Comparing the three baseline models of mid-latitude,

tropical and arctic environments, the performance simulations find that highest precision measurements can be achieved under tropical conditions due to high humidity levels and favourable aerosol backscattering. Relative random errors lower than 0.1% for $H_2O$ are achievable within the first kilometer. The precision for $H_2O$ degrades faster than for HDO with increasing range due to strong absorption leading to low return signals. On the contrary, random uncertainties for the arctic environment are almost one order of magnitude higher due to rather dry conditions in terms of WVMR and low aerosol

content observed at the Eastern Greenland AERONET station of Ittoqqortoormiit. A high sensitivity to seasonal variability of the humidity profile was observed for the arctic model, whereas variations of humidity in the tropics throughout the year are small and thus only slightly affect the expected measurement precision. The simulations also clearly show the influence of aerosols on the performance of DIAL measurements. For all three locations, the precision gain between the low-charge (lowest-decile AOD) and high-charge (highest-decile AOD) aerosol model is roughly one order of magnitude. The presence

of free-tropospheric aerosols, for example due to long-range dust transport in the mid-latitudes and arctic haze or boreal forest fire smoke in the Arctic, leads to significant improvements in the precision at altitudes beyond the atmospheric boundary layer. Adapting the measurement bandwidth, along the line of sight for instance, and of course adapting the integrating time could be envisioned to retrieve nominal performances under these different atmospheric conditions.

### 3.3 Systematic errors

Systematic errors are associated with an uncertainty in the knowledge of atmospheric, spectroscopic, and instrument-related parameters when obtaining the VMR from the measured differential optical depth according to Eq. (4). Expressed in a



general form, errors were estimated by calculating the VMR retrieval sensitivity to a deviation $\delta Y$ from a reference parameter $Y$:

$$\varepsilon_s = \max\left\{\frac{|X(Y) - X(Y \pm \delta Y)|}{X(Y)}\right\} \tag{14}$$

For the case of atmospheric systematic errors, the reference parameter $Y$ used for the VMR retrieval stands for either the vertical pressure or temperature profile of the baseline atmospheric model. The systematic error due to an uncertainty in the knowledge of the temperature profile was calculated for temperature deviations $\delta T$ from the reference profile ranging from $\pm 0.5$ K to $\pm 2$ K. As shown in Fig. 7, this kind of error can lead to a significant contribution to the error budget. The analysis shows that sounding $H_2O$ at the absorption peak is especially sensitive to temperature uncertainties and that a measurement

with the on-line wavelength shifted off the absorption peak (option 2) significantly reduces this bias. Similarly, a pressure deviation $\delta p$ ranging from 0.5 hPa to 2 hPa was used to estimate the error due to an uncertainty in the pressure profile. In this case, wavelength option 2 is more sensitive to such an uncertainty. The resulting bias on the measurement of HDO is found to be negligible. Note the difference between the two options for probing $H_2O$. Shifting the online wavelength off the absorption peak (option 2) results in a noticeable reduction in the temperature error. However, this comes at the expense of

increased pressure error and lower signal-to-noise ratio and thus increased random error for unchanged laser energy, integration time, and bandwidth.

For the case of instrument-related errors, we assume perfect spectral quality of the laser source and estimate only the systematic error arising from the accuracy of the transmitter on- and offline wavelengths. For our estimate we use a laser frequency deviation $\delta f$ ranging from 2.5 MHz to 10 MHz corresponding to wavelength stabilities reliably achievable over

several minutes with our envisioned OPO/OPA approach coupled to a commercial wavemeter, which can suffer thermal drifts of a few MHz over several tens of minutes. The relative wavelength error was calculated according to Eq. (14) by introducing a wavelength detuning $\delta f$ to the on- and offline wavelengths. Due to the narrower absorption line of $H_2O$ at 1982.93 nm, we find that such wavelength detuning results in a much larger error compared to the spectrally larger HDO line. Option 2 for $H_2O$ measurement drastically reduces the wavelength error.

Considering the three mentioned systematic error contributions, option 2 proves to be the preferred wavelength choice with the intention of reducing the systematic error, especially if the temperature profile along the line of sight is not known with accuracy better than $\pm 0.5$ K.

Another systematic error arises for the HDO retrieval from the insufficient knowledge of the optical depth due to a non-negligible $H_2O$ absorption feature at the HDO line at 1982.47 nm. Assuming a relative uncertainty of 1.5% in the VMR

profile of $H_2O$, which is a conservative estimate for the combined systematic error of the $H_2O$ measurement due to temperature, pressure, and wavelength uncertainty, calculations reveal relative errors in the VMR retrieval varying between 0.23% for the arctic model and 0.55% for the tropical model. It should be noted that even for the measurement of $H_2O$ an interference contribution due to higher HDO absorption at the off-line wavelength leads to a bias. However, this error is





relatively small compared to other systematic errors and the achievable random error which justifies the proposed method of

calculating the $H_2O$ VMR with no a-priori knowledge of HDO and then using the obtained profile to correct the differential optical depth of the HDO retrieval according to Eq. (6).

Finally, systematic errors in the VMR retrieval due to uncertainties related to spectroscopic parameters were analyzed by introducing deviations between 1% to 5% to the HITRAN 2016 parameters of line intensity and air-broadened width and deviations of 1% to 10% to the temperature-dependence width coefficient and the pressure shift parameter. The resulting

systematic errors are shown in Fig. 8 for each parameter. Uncertainties in parameters of line intensity and air-broadened width largely contribute to the error budget highlighting the importance of the precise knowledge of these quantities. It should be noted that the assumed uncertainties have a rather demonstrative character as their precise quantification is still the subject of ongoing spectroscopic studies. A summary of the presented systematic errors in the form of an error budget for each of the three atmospheric models is given in Table 2.

**3.4 Precision estimate applied to field campaign data**

In order to complete our previous numerical studies and relate to more realistic atmospheric conditions, we present here the results of performance calculations initialized with observations obtained during the L-WAIVE (Lacustrine-Water vApor Isotope inVentory Experiment) field campaign at the Annecy lake in the French alpine region (Chazette et al., 2020). This experiment was specifically carried out in order to obtain reference profiles that can be used to simulate the WaVIL lidar

vertical profiles. Hence, the data include vertical profiles of pressure and temperature as well as vertical profiles of $H_2O$ and HDO isotopologue concentrations which were obtained by an ultra-light aircraft equipped with an in-situ cavity-ring-down-spectrometer (CRDS) isotope analyzer. As aerosols were present above the planetary boundary layer on 14 June 2019, we chose data acquired from that day, ranging up to an elevation of 2.3 km. To simulate atmospheric conditions during the measurement campaign as realistically as possible, we used aerosol extinction data from the lidar WALI (Weather and

Aerosol Lidar) (Chazette et al., 2014) operated during the L-WAIVE campaign on the same day (see Fig. 8a). The backscatter coefficient was estimated with a lidar ratio of 50 sr and extrapolated to a wavelength of 2 µm using the Angstrom exponent derived from sun-photometer measurements. For the purpose of our simulation study, we do not take into account any measurement uncertainties in the described profiles. Figures 8b and 8c show the expected precision in the DIAL-retrieved isotopic ratio in terms of $\delta D$ depending on detector characteristics and laser energy (calculation based on

wavelength option 1 for $H_2O$). For the commercial InGaAs PIN photodiode the simulations show for the limiting case of 20 mJ laser energy that the uncertainty related to noise is sufficiently low so that the characteristic variations in the experimentally obtained $\delta D$ profile could be fully resolved with the proposed DIAL system. The expected absolute precision for this configuration is well below 5‰ within the first 1.5 km and reaches 10‰ at a range height of 2.3 km. A setup with 10 mJ would deliver an absolute precision of 20‰ at that height. The expected precisions are on the order of or better than

the columnar measurements obtained with other remote sensing techniques deployed from the ground (between 5 and 35‰ for Fourier Transform Infrared Spectrometer and Total Carbon Column Observing Network) or from space (~40‰ for the



Tropospheric Emission Spectrometer and the Infrared Atmospheric Sounding Interferometer, see Table 1 of Risi et al., 2012) but with a much greater resolution on the vertical. On the other hand, the expected precision is roughly 2 to 4 times lower than for in situ airborne CRDS measurements with a similar vertical resolution (see Table 3 of Sodemann et al., 2017).

Simulations performed with the HgCdTe APD indicate extremely promising precision levels over the entire range of under 3‰ and 5‰ (in absolute terms) for 20 mJ and 10 mJ, respectively. It should be noted that the presented profiles represent a rather favourable case since the aerosol backscatter coefficient increases with altitude (due to the presence of an elevated dust layer) which is the contrary to the baseline atmospheric models described in the previous numerical analysis. These simulations incorporating observed $H_2O$ and HDO profiles clearly show the potential of a ground-based DIAL instrument to

measure isotopic mixing ratios with high spatio-temporal resolution in the lower troposphere.

**4 Conclusion**

Probing the troposphere for water isotopologues with high spatio-temporal resolution is of great interest to study processes related to weather and climate, atmospheric radiation, and the hydrological cycle. In this context, the Water Vapor Isotope Lidar (WaVIL), which will measure $H_2O$ and HDO based on the differential absorption technique, is under development.

The spectral window between 1982–1984 nm has been identified to perform such measurements. Indeed, HDO displays sufficiently high ab-sorption lines in this range. Interference with the main isotopologue $H_2O$ is manageable, especially since both species would be measured simultaneously.

We performed a sensitivity analysis and an error budget for this system taking instrument-specific and environmental parameters into account. The numerical analysis included models of mid-latitude, polar, and tropical environments with

realistic aerosol loads derived from the AERONET database extrapolated to the 2 μm spectral region. We showed that the retrieval of $H_2O$ and HDO mixing ratios is possible with relative precisions better than 1% within the atmospheric boundary layer in mid-latitude and tropical conditions, the latter giving rise to the highest precision due to favourable differential absorption. Performance simulations also revealed differences in precision of almost one order of magnitude between the tropical and arctic model. Reduced precision under arctic conditions is due to low water vapour content and reduced aerosol

load. These findings have been obtained for laser pulse energies of 20 mJ, a measurement bandwidth of 1 MHz (150 m range resolution), an integration time of 10 min, and a commercial InGaAs PIN photodiode. As an interesting perspective option, we also investigated the theoretical performance of a state-of-the-art HgCdTe avalanche photodiode featuring a NEP reduced roughly by one order of magnitude. The use of such a detector would relax the requirement on laser energy and integration time and enable high-precision, range-resolved measurement of the isotopic ratio.

An error budget has been performed to outline systematic errors due to uncertainties in atmospheric, spectroscopic, and instrument-related parameters. The $H_2O$ on-line wavelength at 1982.93 nm shows a pronounced temperature sensitivity imposing strict requirements on accurate temperature profiles for the VMR retrieval. This can be mitigated by tuning the on-line wavelength to 1982.97 nm which, however, comes at the cost of slightly increased pressure sensitivity and reduced





differential absorption. Including systematic errors due to inexact spectroscopic parameters in our analysis, we highlighted
the importance of their accurate knowledge for DIAL measurements and the necessity for ongoing spectroscopic studies of
water vapor isotopologues in the two-micrometer region.

Finally, using a measured $H_2O$/HDO profile obtained during the recent L-WAIVE field campaign, our calculations have
shown that sufficient precision in the mixing ratios of $H_2O$ and HDO can be achieved with the presented system parameters
so that characteristic, vertical variations of the isotopic content $\delta D$ can be resolved with the proposed DIAL system, showing
the potential to complement existing methods. Future work will consist of improving our knowledge in the spectroscopy of
HDO in the 1982–1984 nm spectral region and testing the DIAL system in the framework of a forthcoming field campaign.

## Appendix A

Table A1 lists databases and locations used to derive the three atmospheric models discussed in this paper. Available data
from the year of 2019 was used for all locations.

**Table A1. Overview of atmospheric sounding and AERONET sites used to derive an atmosphere model for the sensitivity analysis.**
**For all sites data from 2019 was used. Note that for the arctic station, AERONET photometer products are from February until**
**September.**

|  | Sounding profiles (pressure, temperature, humidity) | AERONET (Level 2 aerosol optical depth) |
|---|---|---|
| **Mid-latitude station:** Paris region, France | Trappes 48.77°N, 2.01°E Météo France data (https://donneespubliques.meteofrance.fr/) | Palaiseau 48.71°N, 2.22°E |
| **Arctic station:** Ittoqqortoormiit, Denmark | Ittoqqortoormiit 70.49°N, 21.95°W University of Wyoming data (http://weather.uwyo.edu/upperair/sounding.html) | Ittoqqortoormiit 70.49°N, 21.95°W February – September 2019 |
| **Tropical station:** La Réunion Island, France | La Réunion (Gillot) 20.89°S, 55.51°E Météo France data (https://donneespubliques.meteofrance.fr/) | La Réunion (St. Denis) 20.90°S, 55.49°E |

## Author contribution

Conceptualization of measurement concept, J.H., J.-B.D., M.R., R.S., A.G., J.-M.M., J.T., P.C. and C.F.; lidar performance
simulator, J.H.; L-WAIVE campaign data curation, H.S., A.S., J.T. and P.C.; writing—original draft preparation, L.R., C.C.,
M.R. and J.H.; writing—review and editing, all authors; project administration, C.F. and C.E.; All authors have read and
agreed to the published version of the manuscript.



**Competing interests**

The authors declare that they have no conflict of interest.

**Acknowledgements**

The authors would like to thank the AERONET network and responsible PIs for sun photometer products (https://aeronet.gsfc.nasa.gov/, last access: 14 November 2020). The authors acknowledge the provision of meteorological sounding data by the University of Wyoming (http://weather.uwyo.edu/upperair/sounding.html, last access: 16 November 2020) and the French national meteorological service Météo-France (https://donneespubliques.meteofrance.fr, last ac-cess: 26 November 2020).

This work was partially funded through WAVIL project: ANR grant Nr. ANR-16-CE01-0009 and has received funding from the European Union's Horizon 2020 research and innovation program under grand agreement Nr. 821868. H.S. and A.S. acknowledge funding obtained through the ERC Consolidator Grant Nr. 773245 (ISLAS).

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



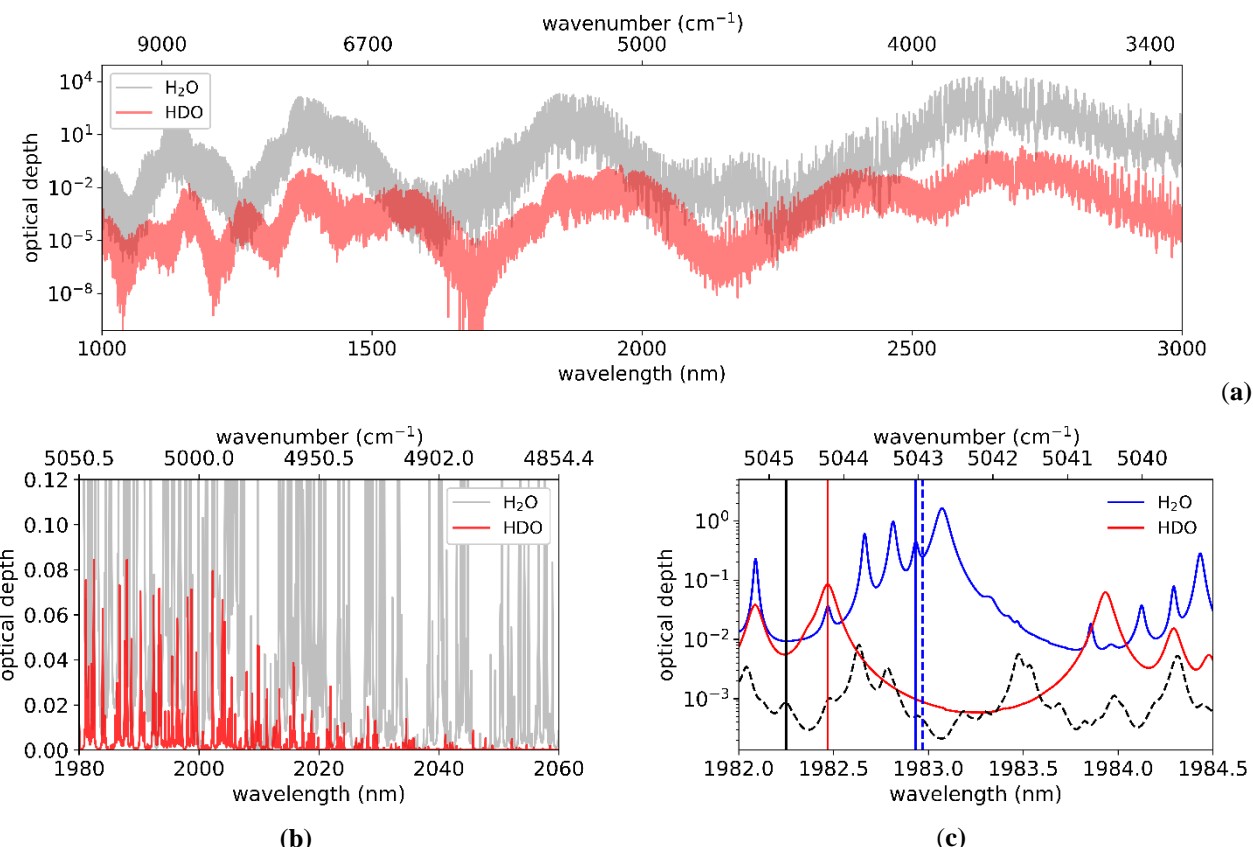

**(a)**

**(b)**

**(c)**

**Figure 1: Optical depth over 1 km for H₂¹⁶O (H₂O) and HD¹⁶O (HDO) with uniform volume mixing ratios of 8400 ppmv and 2.6 ppmv, respectively. (a) Spectral overview between 1 μm and 3 μm; (b) Close-up window for wavelengths around 2 μm with decreasing HDO absorption towards 2.05 μm; (c) Spectral range of interest for simultaneous H₂O and HDO sounding. The dashed black line represents the total optical depth of other species (CO₂, CH₄, N₂O) with their typical atmospheric concentrations. The vertical black line indicates the position of the off-line wavelength. On-line wavelengths are indicated for H₂O (vertical blue line**
**for option 1 at 1982.93 nm, dashed line for option 2 at 1982.97 nm) and HDO (vertical red line). Spectra calculations are based on the HITRAN 2016 database assuming a temperature of 15°C and an atmospheric pressure of 1013.25 hPa.**





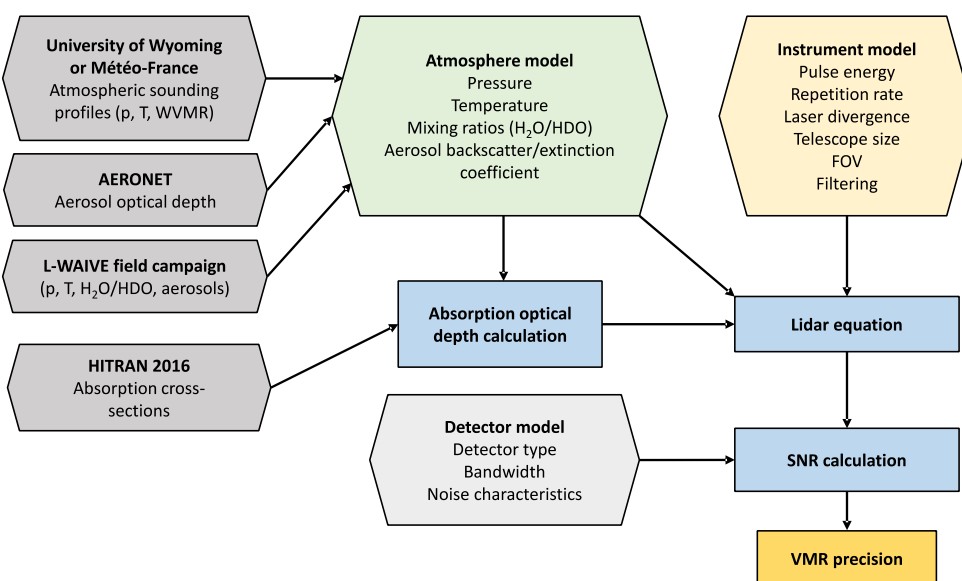

**Figure 2: Block diagram of the DIAL simulator. Input models and databases in hexagons, principal calculations indicated by rectangles. p: pressure, T: temperature, WVMR: water vapor mixing ratio, FOV: telescope field of view. The signal-to-noise ratio (SNR) is used to calculate the statistical random error (precision) of the volume mixing ratio (VMR) of $H_2O$/HDO.**


**Figure 3: Atmosphere models: (a–c) Vertical sounding profiles of pressure, temperature, and water vapor mixing ratio (WVMR). Grey lines indicate monthly averages, solid green line is the yearly average of 2019 (baseline profile). Dotted lines indicate profiles of lowest and highest monthly temperatures and WVMR; (e–f) Model profiles of aerosol extinction coefficient; (g–h) Distribution of the aerosol optical depth at 1983 nm for AERONET level 2.0 data of 2019.**






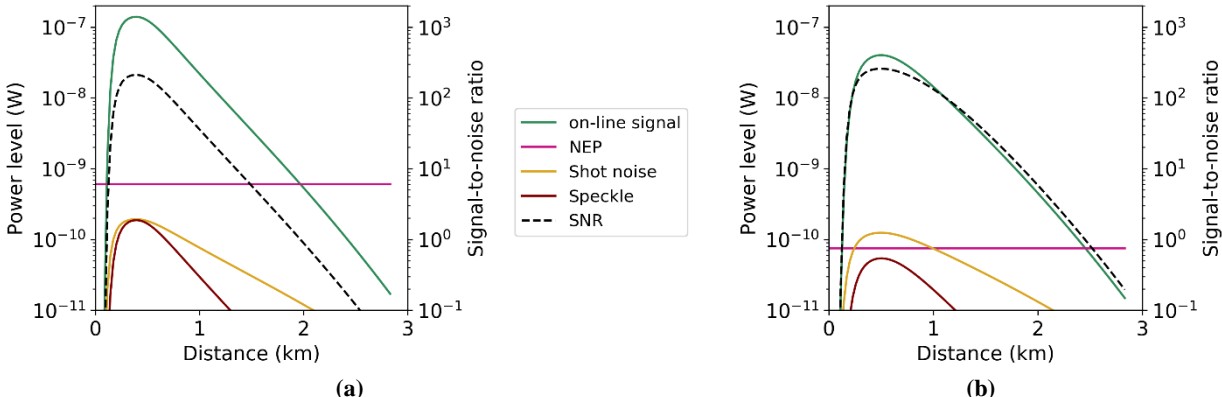

**Figure 4: Received power according to Equation (2) (solid green line) and power-equivalent levels of major noise contributions related to the $H_2O$ on-line signal for a single 20 mJ pulse and resulting signal-to-noise ratio (SNR, dashed black line, right vertical axis) as function of lidar range: (a) InGaAs PIN detector; (b) low-noise HgCdTe APD.**

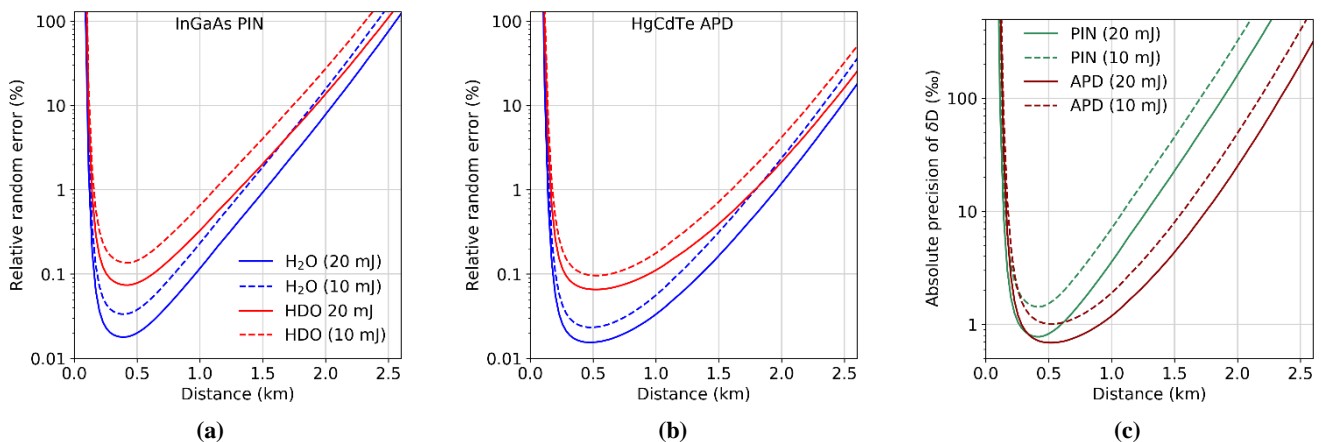

**Figure 5: Expected relative random error on the volume mixing ratio of $H_2O$ and HDO for different pulse energies and detectors: (a) InGaAs PIN detector; (b) HgCdTe APD; (c) Corresponding absolute uncertainty (standard deviation) on $\delta D$ as a function of distance from the lidar instrument. A detection bandwidth of 1 MHz is assumed and signal averaging time is 10 min.**





**Figure 6: Sensitivity with respect to variability of atmospheric parameters: resulting statistical uncertainty for range-resolved DIAL measurement of H₂O (blue, only wavelength option 1) and HDO (red). Simulation parameters: 20 mJ pulse energy, 1 MHz bandwidth, 10 min integration time, InGaAs PIN detector. (a–c) Reference model based on average columns of pressure, temperature, and humidity. Aerosol baseline profile using median AOD assumed; (d–f) Sensitivity to water vapor variability; (g–i) Sensitivity to different aerosol profiles (H₂O); (j–l) Sensitivity to different aerosol profiles (HDO).**





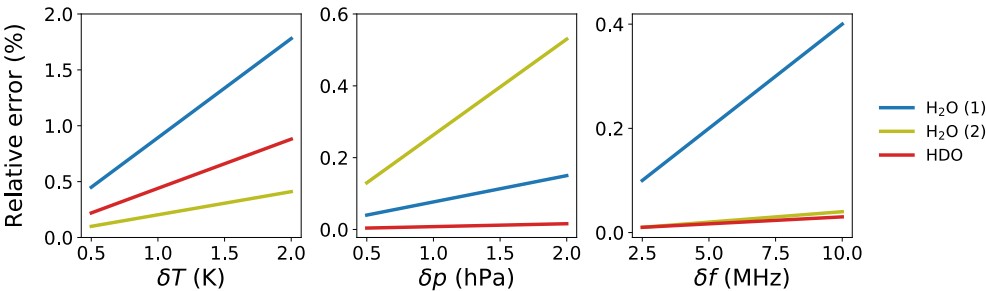

**Figure 7: Maximal relative error in the VMR retrieval (over 3 km range) due to uncertainties in the profiles of temperature ($\delta T$) and atmospheric pressure ($\delta p$) as well as transmitter on- and off-line wavelength ($\delta f$). (1) and (2) stand for the two possible on-line wavelength options for measuring $H_2O$.**

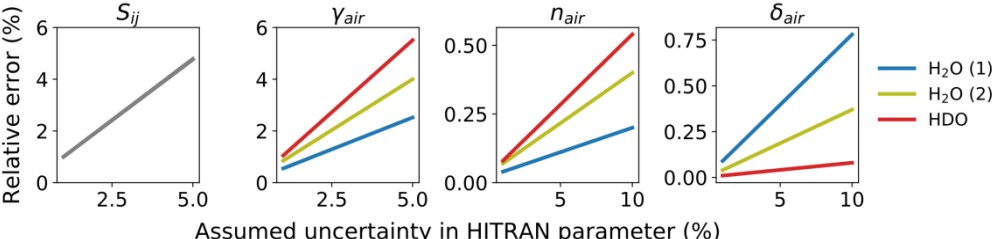

**Figure 8: Maximal relative error in the VMR retrieval (over 3 km range) due to uncertainties in HITRAN parameters. $S_{ij}$: line intensity; $\gamma_{air}$: air-broadened half width; $n_{air}$: coefficient of the temperature dependence of $\gamma_{air}$; $\delta_{air}$: pressure shift.**



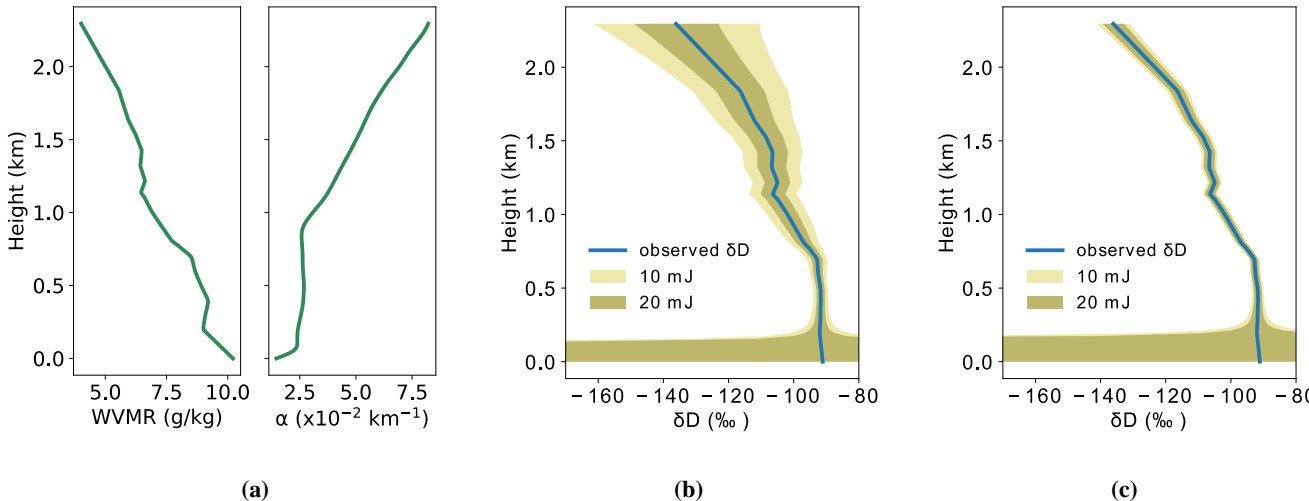

(a)        (b)        (c)

**Figure 9: (a) Experimental profiles of water vapor mixing ratio (WVMR) and aerosol extinction coefficient ($\alpha$) obtained from the L-WAIVE field campaign. Expected precision in the isotopic ratio in terms of $\delta D$ for the InGaAs PIN photodetector (b) and the low-noise HgCdTe avalanche photodiode detector (c). Shaded areas indicate the absolute uncertainty based on random noise in terms of standard deviation for laser energies of 10 mJ and 20 mJ. High uncertainty in the first 200 m is due to the overlap function increasing from zero to unity (see Eq. (2)). Calculations based on a measurement bandwidth of 1 MHz (150 m spatial resolution) and an integration time of 10 min.**

625



**Table 1: DIAL instrument parameters**

| Transmitter | | Receiver | | |
|---|---|---|---|---|
| | | | i) | ii) |
| Energy | 10–20 mJ | Telescope aperture | 40 cm | 40 cm |
| Pulse duration | 10 ns | Detector type | InGaAs PIN | HgCdTe APD |
| Repetition rate | 150 Hz | Detector diameter | 300 µm | 180 µm |
| $\lambda_{on}$ ($H_2^{16}O$) (1) | 1982.93 nm | Field of view (FOV) | 630 µrad | 380 µrad |
| $\lambda_{on}$ ($H_2^{16}O$) (2) | 1982.97 nm | NEP | 600 fW Hz$^{-1/2}$ | 75 fW Hz$^{-1/2}$ |
| $\lambda_{on}$ ($HD^{16}O$) | 1982.47 nm | Bandwidth | 1 MHz | 1 MHz |
| $\lambda_{off}$ | 1982.25 nm | Responsivity: 1.2 AW$^{-1}$ | | Quantum efficiency: 0.8 |
| Divergence | 270 µrad | | | Excess noise factor: 1.2 |

630

**Table 2: Systematic errors for mid-latitude, arctic, and tropical atmospheric models (maximal error over 3 km range). (1) and (2) denote the two wavelength options for H₂O measurement.**

| Parameter | Assumed uncertainty [1] | Maximal relative error $\varepsilon_s$ (%) | | | | | | | | |
|---|---|---|---|---|---|---|---|---|---|---|
| | | *Mid-latitude* | | | *Arctic* | | | *Tropic* | | |
| | | $H_2O$ (1) | $H_2O$ (2) | HDO | $H_2O$ (1) | $H_2O$ (2) | HDO | $H_2O$ (1) | $H_2O$ (2) | HDO |
| Temperature | ±0.5 K | 0.45 | 0.10 | 0.22 | 0.50 | 0.13 | 0.23 | 0.40 | 0.08 | 0.22 |
| Pressure | ±1 hPa | 0.08 | 0.27 | 0.01 | 0.08 | 0.27 | 0.01 | 0.08 | 0.27 | 0.01 |
| VMR of H₂O bias | 1.5% [2] | - | - | 0.38 | - | - | 0.23 | - | - | 0.55 |
| On/off wavelength | 5 MHz | 0.21 | 0.03 | 0.01 | 0.21 | 0.04 | 0.01 | 0.21 | 0.03 | 0.02 |
| *HITRAN 2016 parameters* | | | | | | | | | | |
| Line intensity | 1% | 1.00 | 1.00 | 1.01 | 1.00 | 1.00 | 1.01 | 1.00 | 1.00 | 1.01 |
| Air-broadened width $\gamma_{air}$ | 1% | 0.55 | 0.85 | 1.05 | 0.54 | 0.87 | 1.07 | 0.53 | 0.83 | 1.05 |
| Temperature coefficient of $\gamma_{air}$ | 5% | 0.13 | 0.2 | 0.33 | 0.19 | 0.31 | 0.49 | 0.07 | 0.1 | 0.16 |
| Pressure shift | 5% | 0.43 | 0.12 | 0.03 | 0.44 | 0.11 | 0.04 | 0.41 | 0.14 | 0.03 |
| **Combined (geometric sum)** | | **1.33** | **1.36** | **1.56** | **1.35** | **1.40** | **1.59** | **1.29** | **1.34** | **1.58** |

[1] relative uncertainty if stated in %
[2] conservative estimate of combined systematic error for H₂O measurement

635