# Peer review of "Differential absorption lidar for water vapor isotopologues in the 1.98 µm spectral region: sensitivity analysis with respect to regional atmospheric variability"

_Atmospheric Measurement Techniques, 2021_

## Author Comment (AC1)

[revised manuscript text omitted]

[1] relative uncertainty if stated in %

[2] conservative estimate of combined systematic error for $H_2O$ measurement

**Author's response to reviewer comments**

We thank both reviewers for their helpful comments and suggestions.

705 **Reviewer 1:**

This is an interesting paper on the predicted performance of a theoretical DIAL system to remotely measure water vapor isotopes. The authors have identified a region in the infrared around 1983 nm that shows potential for measurement of both H(16)OH and HD(16)O. Overall this is a very thorough analysis and is well done.

710 General comments:

1) To have a scientific impact, is the desired/required goal to achieve a <1% relative error? Is a specific water vapor isotope abundance precision in permil required?

The better than 1% relative precision for $H_2O$/HDO mixing ratios can be understood as required limit in order to still reach precisions of <10 permil in the isotopic abundance. Since the isotopic abundance can vary by a few tens of permil,

715 precisions greater than that value would be of not much use for observations in a scientific context.

*Modifications made:* Page 1, line 28 in abstract: Specifies why it is important to have relative precision in $H_2O$/HDO of better than 1%

720 2) The laser development needed to bring the proposed instrument to reality will be far from trivial. The modeling effort assumes that the transmitter will have perfect qualities required for DIAL (such as spectral purity) which is a significant assumption. To be fair, the conclusions could be more clear to say that the predicted performance is a best-case scenario because of this assumption.

Indeed, the calculations represent rather optimistic best-case scenarios.

725

*Modifications made:* Page 16, line 487 et seq: Emphasis that calculations are best-case scenario and do not take into account important aspects which are challenging in practice (such as perfect spectral purity).

3) While reading through, the biggest question I had was if the temperature sensitivity of the line strength was included in the
730 model. I think the answer is yes, but it wasn't very clear in the text.

Yes, temperature dependence of the line strengths is included in the model. We added supplementary equations, tables as suggested in the points below.

3.1) Line selection criteria for water vapor DIAL requires choosing lines that (1) avoid interferences, (2) have the appropriate line strength (will provide the optimal optical depth), and (3) are insensitive to temperature as highlighted in Browell et al. 1991 (Browell, E. V., S. Ismail, and B. E. Grossman, 1991: Temperature sensitivity of differential absorption lidar measurements of water vapor in the 720 nm region. Appl. Opt., 30, 1517–1524). Could this line selection consideration be more clearly articulated in section 2.1?

More emphasis was put on the fact that the chosen absorption lines are not temperature independent as ideally preferred for DIAL. This requires the accurate knowledge of the temperature profile, which has to be provided by auxiliary measurements (for example a Raman lidar next to the DIAL).

*Modifications made:* page 4, lines 125-137

3.2) In the sections around Equation 5 would it help to include, or at least mention the temperature dependence in the absorption cross-sections?

Equations for the Voigt line shape as well as the temperature dependence of the line strength have been added.

*Modifications made:* page 6, lines 174 et seq: Equations 6 and 7 added

3.3) The HD(16)O line at 1982.47 nm has a reasonably temperature-insensitive ground state energy of 91 cm^-1 (something closer to 250 cm^-1 would be ideal), but the interference H(16)OH line, directly underneath it, has a ground state energy of 2756 cm^-1. Is not some discussion warranted about how this creates a high sensitivity to temperature? Using the equations in Browell et al. (1991) it appears that the uncertainty in the number density would be up to 5% per K on line center. Unfortunately, the optical depth when tuned to the line center of that HD(16)O line is already less than optimal for DIAL. But should there be an online second option evaluated, moving to the weaker line at 1983.93nm to avoid this even if constraining the measurements to regions with high water vapor concentrations?

It is true that the initially proposed HDO line at 1982.47 nm violates a number of basic DIAL line selection rules. As shown in our error budget, measuring HDO at this line is only possible if 1) the temperature profile is well known (applies to all selected lines, practical implementation via Raman lidar for example) and 2) the distribution of the main isotope is known to correct for the interference bias. In theory this is all possible, but in practice this procedure adds further uncertainties, indeed. As suggested, we included the HDO line at 1983.93 nm into our analysis. Despite slightly reduced optical depth (thus higher random error, but could be offset by longer averaging), this should definitely be the preferred option for HDO.

*Modifications made:*
page 4, line 124: Introduction of the second HDO option at 1983.93 nm;

page 11, line 310: HDO line at 1983.93 nm leads only to slight increase in expected random error

Figure 5 (page 26). Figure has been updated/modified substantially to include second option for HDO. Figure has been split into upper and lower panel for the two different detectors.

3.4) The online for H(16)OH at 1982.93 nm has a ground state of 920 cm^-1 which is still rather large compared to a typical water vapor DIAL system. Would it be helpful if the ground state energies were listed somewhere to help the reader understand that the model is taking these fundamental factors into account?

Table 1 was added in section 2.1 at page 4 listing important spectroscopic parameters for the investigated absorption lines.

4) The larger than typical DIAL errors/sensitivity to atmospheric temperature seems to be sidestepped somewhat by simply constraining the model temperature uncertainty to +/- 0.5ºK. Although there is a sentence in the conclusions about this, it is a significant issue and warrants more discussion. Could the authors suggest how this will be done in practice? For example, if reanalysis data will be needed to reach these levels of certainty, the DIAL measurement would not be useful in real time. Are they expecting another instrument like a Raman lidar to provide this information?

As already mentioned, temperature data has to be provided by auxiliary measurements (for example by radio sounding or a Raman lidar). For a future field campaign, it is intended to employ a Raman lidar next to the here presented DIAL. So indeed, real time measurements would not be possible. Please note that we changed the assumed temperature uncertainty for the error budget in Table 3 from +/-0.5 to +/-1K which is an accuracy more realistically achievable in the tempeture measurement by Raman lidar.

*Modifications made:*

the aspect of having to have an auxiliary temperature measurement was added at multiple locations: p. 4 lines 129 et seq, p. 13 lines 362-363, in the conclusion, p. 16 lines 475 et seq.

5) Was the model using a yearly or seasonal mid-latitude average (high and low)? It was unclear if the proposed instrument is expected to perform well in mid-latitude winter conditions.

The model uses yearly averages as a baseline. Since in mid-latitude winter conditions less water vapor is present in the atmosphere, the instrument is expected to perform less well than in summer conditions. The main effect here is clearly the water vapor content and thus the difference in differential absorption optical depth. When conducting the calculations with fixed water vapor profile but lowest and highest monthly temperature profiles (green dashed lines in Fig. 3 a-c), no significant difference in the achievable precision was found.

6) Was the performance model limited to nighttime only? Was there any solar background modeled?

We used a constant solar background irradiance of 1 W/m²/µm/sr and an optical filter bandwidth of 50 nm for our calculations. This information was missing before and was added to the manuscript. Shot noise due to solar background added to the plots in Fig. 4.

805     *Modifications made:*

page 8, line 229: background irradiance and filter bandwidth specified

Figure 4: Shot noise due to solar background has been added to the graphs

7) What is the expected lowest useful range of the proposed instrument? The plots in Figure 6 indicate measurement would
810     be limite d to approximately >150m above ground level (if 1% error is the goal). However, based on the curve shapes, the full overlap is not achieved until above 500m which could push the minimum range upward. As I'm sure the authors know, very slight differences in online and offline overlap or pointing can result in large systematic errors when pushing too far into the incomplete overlap region. Could these limitations be discussed a bit more clearly so the science community has realistic expectations of the proposed DIAL instrument?

815     Indeed, the plots in Figs. 5 and 6 suggest that high measurement performance can be already achieved onwards from 150 m or so. We added the aspect of incomplete overlap leading to biases due to laser beam pointing. Please note that all curve shapes in Figs. 4, 5, 6 and 9 have slightly changed due to an error we identified in the submitted version concerning the overlap. Now with the correct parameters used, the lidar instrument reaches full overlap between 450-500 m. In the submitted version it was closer to 800 m due to an error. As suggested by the reviewer, we included a sentence stating the
820     lowest useful range to be around 500 m.

*Modifications made:*

Figs. 4, 5, 6 and 9 have been updated due to an error in the overlap function of the submitted version

page 11, lines 318 et seq: specification of the lowest useful range due to incomplete overlap at ranges <0.5 km

825

Specific comments:

1) line 46. Does it have to be a high-power laser transmitter?

High-power laser transmitter was simply the wrong word. It is the laser energy which has to be sufficient (mJ level) for range-resolved DIAL measurements.

830     *Modifications made:* now page 2, line 47: word "high-power" has been deleted

2) line 101. "… with and out of a gas absorption feature.." is unclear

*Modifications made:* page 4, line 101…formulated differently

    3)   line 155 "two wavelength in (lambda_on) and out of (lambda_off) coincidence" is a bit awkward in English

*Modifications made:* page 6, line 158…formulated differently

4)   line 157, the lasers also need to be sufficiently close in wavelength (not just time)

*Modifications made:* page 6, line 163…"close in wavelength and in time"

5)   line 166, Why use a mixing ratio?   A DIAL measures the number density of the absorbing molecule. When converting to a mixing ratio one has to assume a pressure and thereby increase the uncertainty.

For the scientific context, mixing ratios are required.

    6)   In section 3.4 the precision estimate is referring to figure 9, yet the text says figure 8 (in multiple locations)

Wrong figure references have been corrected.

7)   The x-axis labels in figure 9 seem to be incorrect?  The text at line 397 says "The expected absolute precision for this configuration is well below 5‰ within the first 1.5 km and reaches 10‰ at a range height of 2.3 km"  but the labels

        don't match so it makes it hard to evaluate what is going on.

The idea of Figure 9 b/c is to visualize the hypothetical precision in the form of an error band (shaded areas) around the dD profile obtained in situ during the field campaign. It is true that it is hardly possible to read the exact precision values from the figure. That's why they have been given in a sentence afterwards. We tried to be more clear in the corrected version (p.15 lines 428 et seq)

    8)   line 415 "Indeed, HDO displays sufficiently high ab-sorption [sic] lines in this range" Suggest adding a qualifier here such as "for measuring the lower 1km of the atmosphere with better than 1% relative error in the mid-latitudes and regions with higher water vapor concentrations."

The suggested qualifier was added to be more concise (p. 15, line 455)

    **Reviewer 2:**

Hamperl et al. presents a theoretical analysis and performance evaluation of a DIAL system to measure vertical profiles of water vapor $H2(16)O$ and its isotopologue $HD(16)O$. The paper is well written and detailed. I recommend it to be published after the following comments are addressed:

    General comments:

1)   I understand the authors decided to exclude the laser linewidth from the analysis, nevertheless I think it would be good if they at least provide a first order estimate of its impact.

An error estimation was provided by introducing the concept of effective absorption cross section for taking into account the finite bandwidth of the laser transmitter. The additional error is found to be in the order of 0.1% for the narrowest H2O line.

870

*Modifications made:* page 13, line 385 et seq: introduction of effective cross sections to calculate numerically the bias due to a laser line width of 50 MHz

2) Is the 'efficiency' of the receiver optics (Tr in Eq. 2) assumed to be 1? If so, is that a reasonable assumption?

875    The efficiency is assumed to be 0.5. We added this specification (p. 6, line 159).

3) The authors include the effect of solar background in Eq. 10, but there is no further discussion regarding its impact on the instrument performance (and the optical filter bandwidth is not included in Table 1).

We used a constant solar background irradiance of 1 W/m²/µm/sr and an optical filter bandwidth of 50 nm for our calculations.

880    This information was missing before and was added to the manuscript.

*Modifications made*: page 8, line 229: background irradiance and filter bandwidth specified

4) As the previous reviewer pointed out, it would be nice to have a more detailed analysis of temperature sensitivity of the

885        line strengths and its impact of the overall retrieval uncertainty.

As suggested, more details on the temperature dependence of the line strengths have been added to show that this aspect is taken into account by the calculations.

please see Reviewer 1, point 3)

890    5) Have you considered exploiting an absorption line with strong temperature dependence to try to retrieve temperature simultaneously? I'm unsure if a reasonable uncertainty can be achieved, but it might be worth exploring it.

We have so far not considered this idea. In the case of a true simultaneous $H_2O$ + temperature measurement, a multi-wavelength switching scheme would have to be realized, which is possible but technically more challenging. And since in that case the repetition rate for each wavelength decreases, the SNR will suffer for unchanged integration time. As shown in

895    the paper, even with only a two-wavelength switching there are conditions under which the instrument performs rather poorly. It is an interesting idea to study by further calculations, but in practice we are not there yet…

900

**Additional remarks**

1) As already mentioned in the response to reviewer 1, the calculations of the originally submitted version contained an error in the overlap function. This has been corrected now and affects Figs. 4, 5, 6 and 9. Updated figures are underlined in red.

2) Please note that the considered field of view for the detector option 2 was changed (see Table 2) to the same value as for detector 1. (p. 7, lines 205 et seq). The FOV angle stated in the original manuscript would be too small in practice and heavily degrade the overlap function.

3) Table 3 has been updated. The assumed temperature uncertainty was changed from 0.5 to 1K. The assumed $H_2O$ bias for HDO (1) was changed to relative uncertainty of 2% for a more conservative estimate. The wavelength errors for $H_2O$ (1) were erroneous and have been corrected.

4) Figure 1c was updated to include second option for HDO at 1983.93 nm.

5) Figures 7 and 8 were updated to include second option for HDO at 1983.93 nm.

6) Line 214: The number of laser shots per wavelength has to be 45.000, not 30.000.